# Piperazine Derivative Stabilizes Actin Filaments in Primary Fibroblasts and Binds G-Actin In Silico

**Nikita Zernov** [1], **Viktor Ghamaryan** [2], **Ani Makichyan** [2], **Daria Melenteva** [1], **Lernik Hunanyan** [2] **and Elena Popugaeva** [1,*]

1   Laboratory of Molecular Neurodegeneration, Peter the Great St. Petersburg Polytechnic University, 195251 St. Petersburg, Russia
2   Laboratory of Structural Bioinformatics, Institute of Biomedicine and Pharmacy, Russian-Armenian University, Yerevan 0051, Armenia
*   Correspondence: lena.popugaeva@gmail.com

**Abstract:** Alzheimer's disease (AD) is characterized by synaptic dysfunction, which is expressed through the loss of dendritic spines and changes in their morphology. Pharmacological compounds that are able to protect spines in the AD brain are suggested to be novel drugs that would be able to slow down the disease progression. We have recently shown that a positive modulator of transient receptor potential cation channel subfamily C member 6 (TRPC6), the compound N-(2-chlorophenyl)-2-(4-phenylpiperazine-1-yl) acetamide (51164), causes the upregulation of postsynaptic neuronal store-operated calcium entry, maintains mushroom spine percentage, and recovers synaptic plasticity in amyloidogenic mouse models of Alzheimer's disease. Here, using confocal microscopy and calcium imaging methods, we present the experimental data indicating that 51164 possesses an alternative mechanism of action. We demonstrated that 51164 can increase the mushroom spine percentage in neurons with the downregulated activity of TRPC6-dependent neuronal store-operated calcium entry. Moreover, we report the binding of 51164 to G-actin in silico. We observed that 51164 interacts with Lys 336, Asp157, and Ser14 of G-actin, amino acids involved in the stabilization/polymerization of the G-actin structure. We showed that interactions of 51164 with G-actin are much stronger in comparison to the well-characterized F-actin stabilizing and polymerizing drug, jasplakinolide. The obtained results suggest an alternative protective mechanism of 51164 that is related to the preservation of actin filaments in vitro.

**Keywords:** piperazine; actin; neuroprotection

## 1. Introduction

Alzheimer's disease (AD) is the most common cause of dementia accounting for an estimated 60% to 80% of cases [1]. The pathologic hallmarks of AD are the formation of extracellular amyloid β peptide (Aβ) plaques outside neurons in the brain and twisted strands of hyperphosphorylated tau protein (tangles) inside neurons [2]. These changes are accompanied by the loss of synapses and neuronal degeneration.

The loss of synapses indicates a reduction in contacts between neurons. Connections between two neurons in most excitatory synapses usually occur through the axon of one neuron and the dendritic spine of another neuron. The greater the dendritic spine head volume, the stronger the synapse. Spines with the largest head volumes are mushroom spines, have thin necks and bulbous heads, and are thought to be cellular indicators of memory storage [3,4]. The elimination of mushroom spine fractions was shown in translational models of AD [5–7] and conditions of amyloid toxicity [8]. Moreover, the reduction in mushroom spines underlies the memory loss observed in AD patients [9,10].

Postsynaptic neuronal store-operated calcium entry (nSOCE) is needed to stabilize mushroom spines [6,8,11–13]. We have previously shown that transient receptor potential cation channel subfamily C member 6 (TRPC6) is a key regulator of nSOCE in hippocampal

neurons [6–8,13]. One of the TRPC6 agonists, N-(2-chlorophenyl)-2-(4-phenylpiperazine-1-yl) acetamide (51164), restores nSOCE, maintains mushroom spine percentage, and recovers synaptic plasticity in amyloidogenic mouse models of AD under conditions of amyloid toxicity [13]. We recently revealed that CaMKIIβ plays a role in regulating nSOCE activity in primary hippocampal culture [14]. In this article, we found that 51164 is unable to upregulate nSOCE under CaMKIIβ knockdown conditions. However, we found that 51164 can maintain the percentage of mushroom spines in the hippocampal culture with a CaMKIIβ knockdown. We suggest that 51164 demonstrates an alternative to the TRPC6–nSOCE mechanism of action. We assumed that 51164 can directly affect the stabilization of actin. In the current paper, we obtained in vitro and in silico experimental data indicating that 51164 can interact with actin.

## 2. Materials and Methods

### 2.1. Chemical Compounds

N-(2-chlorophenyl)-2-(4-phenylpiperazin-1-yl)acetamide (compound 51164 was obtained from the public chemical library InterBioScreen (Chernogolovka, Russia). Cytochalasin D was obtained from Tocris (Tocris, Bristol, UK, #1233).

### 2.2. Mice

Albino inbred mice (FVB/NJ) were obtained from Jackson Laboratory (Jackson Laboratory, Bar Harbor, ME; Stock No: 001800) and used as a source of fibroblasts and brain tissue.

### 2.3. Plasmids

The pCSCMV:tdTomato plasmid was a gift from Gerhart Ryffel (Addgene, Watertown, MA, USA; #30530) [15]. Venus-CaMKIIα was a gift from Steven Vogel (Addgene, Watertown, MA, USA; #29428) [16]. GFP-C1-CAMKIIbeta was a gift from Tobias Meyer (Addgene, Watertown, MA, USA; #21227) [17]. CaMKIIβshRNA plasmid was obtained from Santa Cruz Biotechnology (Santa Cruz, CA, USA; #sc-38951-SH). Control short hairpin RNA interference was obtained from Sigma (Sigma-Aldrich, St Louis, MO, USA; #SHC002). The efficiency and specificity of CaMKIIβ knockdown were checked previously [14].

### 2.4. Primary Hippocampal Neuronal Cultures

Primary hippocampal neuronal cell cultures were prepared by a previously described protocol [6,8,12,14]. Briefly, hippocampal tissue was taken from postnatal days 0–2 FVB/NJ mice, dissected in ice-cold buffer (1% 10× CMF-HBSS (Gibco, Grand Island, NY, USA; #14185), 1% Pen Strep (Gibco, Grand Island, NY, USA; #15140), 16 mM HEPES (Sigma, St Louis, MO, USA; #H3375), 10 mM NaHCO$_3$ (Sigma, St Louis, MO, USA; #S5761); pH = 7.2) dissociated by trituration with 5 mg/mL DNase I solution (Sigma, St Louis, MO, USA; #DN-25) after digestion with papain solution (Worthington, Columbus, OH, USA; #LK003176) for 30 min at 37 °C. Cells were plated on a poly-D-lysine-coated (Sigma, St Louis, MO, USA; #P0899) 24-well culture plate on 12 mm glass coverslips (Thermo Scientific, Braunschweig, Germany; #CB00120RA1). The culture medium consisted of neurobasal (Gibco, Grand Island, NY, USA; #10888) medium supplemented with 1% FBS (Gibco, Grand Island, NY, USA; #10500), 2% 50xB27 (Gibco, Grand Island, NY, USA; #17504), and 0.05 mM L-Glutamine (Gibco, Grand Island, NY, USA; #250030). The cultures were maintained at 37 °C in a 5% CO$_2$ incubator.

### 2.5. Calcium Phosphate Transfection of Primary Hippocampal Cultures

Primary hippocampal neurons were transfected using the calcium phosphate method as previously described [6] with alterations that were described earlier [14]. The transfection kit was obtained from Clontech (TAKARA Biotechnology, Mountain View, CA, USA; #631312).

### 2.6. Dendritic Spine Analysis in Primary Hippocampal Neuronal Cultures

For the assessment of synapse morphology, hippocampal cultures were co-transfected with the TD-tomato plasmid and shCaMKIIβ or shControl plasmids in a 1:1 ratio at DIV7 using the calcium phosphate method and then fixed with 4% paraformaldehyde in PBS, pH 7.4, at DIV14–16. Cells were incubated with 100 nM of 51164 or an equal volume of DMSO for 24 h before fixation. Confocal microscopy parameters and morphological analysis were as previously described [14]. Briefly, a Z-stack of 8–10 optical sections with a 0.2 μm interval was captured using a 100 × lens (UPlanSApo, 100×/1.40 Oil, OLYMPUS, Tokyo, Japan) with a confocal microscope (Thorlabs, Newton, NJ, USA). Each image was captured at 2048 × 2048 pixels with a maximum resolution of 0.05 μm/pixel. At least seven transfected neurons of each group from two independent experiments were used for quantitative analysis. Morphological analysis of dendritic spines was performed by using the NeuronStudio software package [18] as previously described [6,14].

### 2.7. Calcium Imaging

Calcium imaging was performed as previously described [14] using genetically encoded calcium indicator GCamp5.3 one week after transfection. Briefly, glasses with neurons were transferred to the recording chamber of an Olympus IX73 confocal microscope with a 40× lens (LUMPlanFL N, 40 ×/0.80 Water, OLYMPUS, Tokyo Japan) equipped with a fiber-coupled 475 nm LED (Prizmatix, Holon, Israel; UHP-T-475-SR). Images were collected every 2 s with the sCMOS camera (Zyla 4.2P-USB3, Andor, UK) and analyzed with the Micro-Manager 2.0 software (Vale Lab, UCSF, San Francisco, CA, USA).

Cells were incubated in $Ca^{2+}$-free ACSF (140 mM NaCl, 5 mM KCl, 10 mM HEPES, 1 mM MgCl2, 100 μM EGTA) to record the basal fluorescent signals. Then, ACSF was replaced by 300 μL $Ca^{2+}$-free ACSF with $Ca^{2+}$ channel blockers (10 μM D-Ap5 (Tocris, Bristol, UK, #0106), 50 μM nifedipine (Tocris, Bristol, UK, #1075), 10 μM CNQX (Tocris, Bristol, UK, #0190), 1 μM tetrodotoxin (TTX) (Tocris, Bristol, UK, #1078), and 1 μM thapsigargin (Tg) (Tocris, Bristol, UK, #1138)) for the same time for all of the experiments. Then, 3 μL of 200 mM $CaCl_2$ was applied so that the resulting $CaCl_2$ concentration in the cuvette was 2 mM.

Analysis of the data was performed using ImageJ software. The ROI used in the image analysis was chosen to correspond to single spines. At least four co-transfected neurons (each neuron was taken from a separate glass) of each experimental group from two independent experiments were used for quantitative analysis. Independent experiments mean two different primary hippocampal cultures that were grown independently with a time gap of at least four days.

### 2.8. Fibroblast Culture

Fibroblasts were isolated from the tails of 1- to 3-day-old FVB mice. Tails were minced into little pieces in sterile ice-cold dissection buffer (1% 10× CMF-HBSS (Gibco, Grand Island, NY, USA; #14185), 1% Pen Strep (Gibco, Grand Island, NY, USA; #15140)). After, it was digested with papain solution (Worthington, Columbus, OH, USA; #LK003176) for 30 min at 37 °C, then twice triturated with 5 mg/mL DNase I solution (Sigma, St Louis, MO, USA; #DN-25). The cells were grown in DMEM (Gibco, Grand Island, NY, USA; #41965) medium supplemented with 10% FBS (Gibco, Grand Island, NY, USA; #10500), 1% PEST, (Gibco, Grand Island, NY, USA; #15140), 1% sodium pyruvate (Gibco, Grand Island, NY, USA; #11360), 1% MEM NEAA (Minimum Essential Medium Non-Essential Amino Acids, Gibco, Grand Island, NY, USA; #11140) at 37 °C in a humidified 5% $CO_2$ incubator under standard conditions. Fibroblasts were plated in a 24-well culture plate on 12 mm glass coverslips (Thermo Scientific, Braunschweig, Germany; #CB00120RA1) precoated with 0.1 mg/mL poly-d-lysine (Sigma, St Louis, MO, USA; #P0899) in the third passage. At DIV2, immediately after 15 min incubation with 1 μM of 51164 and/or 2.5 μg/mL cytochalasin D (Tocris, Bristol, UK, #1233), the cells were washed with PBS, fixed with 4% paraformaldehyde in PBS, pH 7.3, for 10–15 min at room temperature and permeabilized with 0.1% Triton X-100 in PBS for 5 min. After that, the cells were

stained with 3–5 µg/mL rhodamine-phalloidin (Invitrogen, Carlsbad, CA, USA; #R415) for 10 min at room temperature. Confocal microscopy of the preparations was performed using a confocal microscope (Thorlabs) at 40× magnification (LUMPlanFL N, 40×/0.80 W, OLYMPUS, Tokyo, Japan) with a resolution of 0.3 µm/pixel using ThorImgLS1 software 5 (Newton, NJ, USA). The size of the images was 1024 × 1024 pixels.

### 2.9. Analysis of F-Actin Structure

The measurement of the mean grey values was performed with the ImageJ software. The mean grey value is the sum of the gray values of all the pixels in the cell area divided by the number of pixels. Mean grey values allow us to evaluate the changes in the mass of F-actin.

Fractal dimension analysis was performed with the box-counting method as previously described [19–21]. To calculate the fractal dimension, an image must be split into foursquare boxes with a side length of e, and then the number of boxes $N(e)$ covering any part of the object is calculated. During the next step, the box size is reduced and calculation is performed again; all of these steps are repeated until $e \to 0$. The box-counting fractal dimension ($D$) is defined as:

$$D = \lim_{e \to 0} \frac{\log N(e)}{\log \frac{1}{e}}$$

The fractal dimension was calculated in ImageJ using the FracLac plugin. Before the analysis, it is necessary to select the cell of interest and remove all other objects from the image. The further image must be binarized and only then is FracLac applied with the following values: the number of grid positions was 4, minimum box size = 0 pixels, and maximum box size = 45% of the image. The measurement of the fractal dimension helped us to detect the F-actin reorganization.

### 2.10. In Vitro Statistical Analysis

The in vitro results are presented as mean ± SEM. Normal distribution was checked by the Shapiro–Wilk test. Statistical analysis was performed using the Kruskal–Wallis test following Dunn's multiple comparisons test or two-way ANOVA following Sidak's multiple comparisons test. The $p$ values are indicated in the text and figure legends as appropriate.

### 2.11. Molecular Models Selection

A three-dimensional molecular model of 51164 was created and optimized by using MM2 force fields [22] using the program Chem program office v. 13.057 [23]. The molecular model of G-actin was taken from www.UniProt.org (accessed on 10 September 2022), with the KB identification number of P68135. Jasplakinolide stored in the PubChem database (identification number CID: 6436289) was taken as a control compound showing a positive modulating effect on G -actin.

### 2.12. Molecular Docking

Autodock Vina was chosen as a program for molecular docking [24]. Even though this program works according to the "hard docking" type, in which the maximum possible number of degrees of freedom for the ligand is calculated, and the target is retarded, the predictive ability is quite high [25]. As a methodology for finding the best conformation ligand on the target surface, "blind docking" was applied. This approach is used in cases where the structural and functional characteristics of the target are not always known or there are several functional binding sites on the target surface, leading to target modulation [26].

The initial parameters of molecular docking are given in Table 1.

**Table 1.** The initial parameters of molecular docking.

| N | Parameter | Meaning |
|---|---|---|
|  | Quantity primary conformers | 20 |
| 1 | Exhaustiveness | 200 |
| 2 | Repeatability experiment | 5 |
| 3 | Volume virtual boxing | 96314 Å$^3$ |
| 4 | RMS deviation | ≤2 Å |

### 2.13. Molecular Dynamics

The molecular dynamics of a complex was carried out using GROMACS version 3.3.1 using the force field CHARMM 36 m. The choice of this force field was due to the possibility of describing a wide range of chemical groups and atoms that are part of both biomacromolecules and small molecules [27]. The solvation of the system was carried out using water molecules of the TIP3 P type, which was used with CHARMM 36, thus improving the conformational space [28]. The amount of water and ions was 26,309 molecules. A cube with a volume of 869.677 nm$^3$ was chosen as a spatial box, where each atom in the system was located at a distance of at least 3 Å from the virtual box wall. The simulation time for each system was 10 ns, involving NVT and NPT ensembles, at a temperature of 300 K and a pressure of 1 atmosphere, using a mesh Ewald (PME) method [29]. The coordinates of all atoms were recorded every 2 ps. Relative ligand–target binding energies were calculated using the g energy module in GROMACS 3.3.1 using molecular mechanics and a continuum solvent model. The output files were analyzed involving XMGRACE Version 5.1.19 [30]. The calculation criteria for the interaction radius were calculated according to the standard: the length of hydrogen bonds was 3.4 Å, the length of Coulomb interactions was 9 Å, and the length of van der Waals interactions was 14 Å. Video files (Video S1: «Jasplakinolide with G_actin (violet)»; Video S2: 51164 with G_actin (red).) of molecular dynamic studies of 51164 and jasplakinolide can be found in the Supplementary Materials.

### 2.14. Conformation Analysis and Interaction Visualization

To determine the amino acid residues involved in the process of complex formation, we built conformational interaction maps for the studied compounds with G-actin. Visualization of the results of molecular docking was performed using BIOVIA discovery Studio software v.20.1.0.19295. The VMD program was also used to visualize the results of molecular dynamics [31]. Calculation and visualization of the biophysical indicators of the complexation ligand–target was carried out based on the Gromacs software using the WYSIWYG 2D plotting module for Unix-like Grace operating systems [30].

### 2.15. In Silico Clustering and Statistical Analysis

The process of clustering the results of molecular docking was carried out using a program created based on the FOREL algorithm in the Python environment. This program allowed us to automate the multi-stage process of selecting, preparing, and visualizing the best conformers that met the selection criteria and analyzed the results of molecular docking. Statistical analysis of the results of the study was carried out on the basis of the complex application of standard statistical methods including the calculation of the standard deviations, mean values, and standard mean errors.

### 2.16. Constant Binding Calculation

The binding constant during complex formation was calculated using the Poisson–Boltzmann equation:

$$\Delta Gexp = -RT \ln \frac{1}{K}$$

$\Delta Gexp$ is the total energy of interaction; $R$ is the gas constant; $T$ is the absolute temperature; $K$ is the binding constant [32].

## 3. Results

### 3.1. 51164 Does Not Upregulate Decreased SOCE in Postsynaptic Spines of Primary Hippocampal Neurons with CaMKIIβ Knockdown

We have recently found that the knockdown of CaMKIIβ causes a decrease in nSOCE in dendritic spines in primary hippocampal cultures [14]. Previously, we have shown that 51164 upregulates TRPC6–nSOCE in postsynaptic spines in amyloid toxicity conditions [13]. To elucidate whether CaMKIIβ is necessary to support 51164 dependent upregulation of nSOCE in postsynaptic spines, we performed calcium imaging experiments in primary hippocampal cultures co-transfected with GCamp5.3 and shCaMKIIβ or GCamp5.3 and shControl plasmids. The culture was transfected at DIV7. At DIV13, the cells were incubated with 100 nM of 51164 or an equal volume of DMSO for 24 h. The imaging was carried out at DIV14. 51164 or DMSO were present during the $Ca^{2+}$ imaging procedure in the corresponding experimental groups.

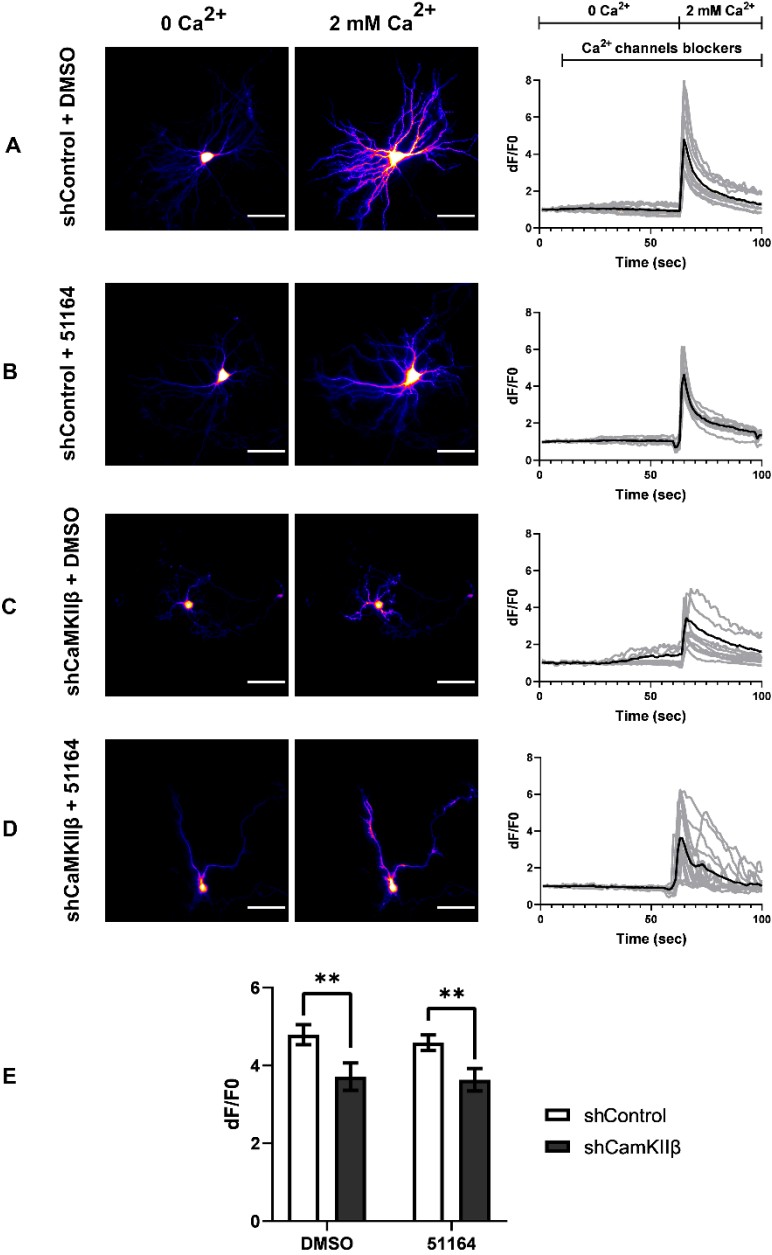

**Figure 1.** 51164 does not rescue a decrease in hippocampal postsynaptic nSOCE amplitude in conditions of CaMKIIβ knockdown. (**A–D**) Representative images of primary hippocampal neurons in the absence ($0Ca^{2+}$) or presence (2 mM $Ca^{2+}$) of $Ca^{2+}$ and time courses of GCaMP5.3 relative fluore-

scence signal changes (F/F0) in the individual dendritic spines. The presence of $Ca^{2+}$ channel blockers (Tg, TTX, Nifedipine, D-AP5, CNQX) and the time of extracellular $Ca^{2+}$ addition are indicated above the traces. Individual spine (gray) and average (black) fluorescence traces are shown for each experimental group. Neurons were co-transfected with either the shControl + GCamp5.3(shControl) or shCaMKIIβ + GCamp5.3 (shCaMKIIβ) plasmids. Results are shown for the neurons that were incubated for 24 h with 100 nM of an equal volume of DMSO (**A**,**C**) or 51164 (**B**,**D**). 51164 (100 nM) was also added to the Ca2+ channel blocker solution during imaging (**B**,**D**). Scale bar = 100 μm. (**E**) Average nSOCE spine peak amplitude was shown for each group of cells. The mean F/F0 peak amplitude signals for each group are presented as the mean $\pm$ SEM (n $\geq$ 100 spines from two independent experiments). Normal distribution was checked by the Shapiro–Wilk test. Statistical analysis was performed using the Kruskal–Wallis test following Dunn's multiple comparisons test, ** $p < 0.01$.

In agreement with our previous result [14], we revealed that nSOCE was downregulated in spines in the absence of CaMKIIβ (Figure 1A,C). There was no difference in the peak amplitude of nSOCE in the shCaMKIIβ + DMSO group versus shCaMKIIβ + 51164 group, $p > 0.9999$, in the Kruskal–Wallis test following Dunn's multiple comparisons test (Figure 1C–E).

Therefore, we ascertained that 51164 does not upregulate decreased nSOCE in the postsynaptic spines of primary hippocampal neurons in the conditions of CaMKIIβ knockdown, indicating that the presence of CaMKIIβ is essential for 51164 mediated postsynaptic calcium entry via nSOCE.

### 3.2. 51164 Recovers Mushroom Spine Percentage in CaMKIIβ Knockdown Hippocampal Cultures

Previously, we have shown that CaMKIIβ knockdown decreases the mushroom spine percentage (% MS) in primary hippocampal neurons, most likely as a consequence of a decrease in postsynaptic nSOCE [14]. It is important to note that the CaMKIIβ knockdown in neurons is not used as a model of AD. In the current study, we reproduced our results and observed that shRNA-mediated CaMKIIβ knockdown caused a decrease in the mushroom spine percentage from 31.8% $\pm$ 1.5% to 18.8% $\pm$ 1.5% (Figure 2A,B). Recently, we proposed that the synaptoprotective effect of 51164 is related to postsynaptic TRPC6-dependent nSOCE upregulation in amyloidogenic models of AD [13]. 51164 did not restore nSOCE amplitude in the absence of CaMKIIβ (Figure 1E), thus we were expecting that 51164 would not recover the mushroom spine percentage in conditions of CaMKIIβ knockdown. Surprisingly, we observed that 24-h incubation in the presence of 100 nM of 51164 increased the mushroom spine percentage in primary hippocampal neurons with a CaMKIIβ knockdown [% MS in the shCaMKIIβ + DMSO group was 18.8% $\pm$ 1.5% in comparison with % MS in the shCaMKIIβ + 51164 (100 nM) group was 33.3% $\pm$ 1.9%, *** $p < 0.0001$, two-way ANOVA following Sidak's multiple comparisons test, n $\geq$ 12 neurons from two independent cultures] (Figure 2). The obtained results demonstrate that 51164 is able to shift the proportion of spines toward the mushroom spines in the CaMKIIbeta knockdown neurons. We speculate that the observed effect might be an alternative synaptoprotective effect that does not include TRPC6-dependent nSOCE.

If not, the nSOCE dependent enlargement of the mushroom spine volume drives the 51164-mediated synaptoprotective effect, then what mechanism provides 51164-dependent neuroprotective properties in primary hippocampal neurons in the absence of CaMKIIβ?

The dynamic cytoskeleton of spines consists of actin filaments [33]. The two leading roles of actin in mature spines are the stabilization of postsynaptic proteins [34] and modulation of the spine shape in response to stimulation [35–37]. The PSD fraction contains a lot of actin-binding proteins such as CaMKIIβ, neurabin-I, drebrin A, etc. [38]. Downregulation of these proteins reduces the formation and maturation of dendritic spines [14,39–41]. Jasplakinolide is a peptide toxin that binds and promotes actin polymerization and thereby stabilizes F-actin. Recently, it has been shown that the application of jasplakinolide to the primary sensory axon preserved the F-actin structure and protected axons from degen-

eration [42]. We hypothesized that 51164 could also bind/stabilize F-actin and therefore provide neuroprotective properties, as demonstrated in Figure 2.

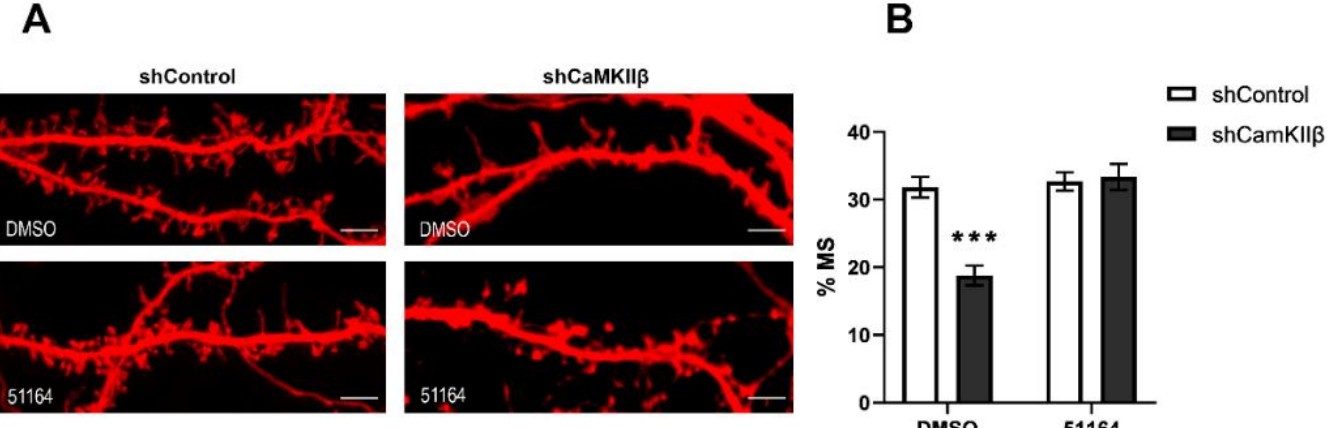

**Figure 2.** 51164 rescues mushroom spine loss in hippocampal neurons with CaMKIIβ knockdown. (**A**) Representative confocal images of dendritic spines in control neurons (shControl, co-transfected with cherry, and shControl plasmids) and in neurons with CaMKIIβ knockdown (shCaMKIIβ, co-transfected with cherry and shCaMKIIβ plasmids). The results are shown for the control conditions (DMSO) and neurons treated with 100 nM 51164 for 24 h (51164). Scale bar = 8 μm. (**B**) Bar chart of mean percentages of mushroom spines (% MS). Results are presented as the mean ± SEM, n (neurons) = 7–10 per group from one experiment, the experiment was repeated twice. Normal distribution was checked by the Shapiro–Wilk test. Statistical analysis was performed using two-way ANOVA following Sidak's multiple comparisons test, *** $p < 0.0001$.

### 3.3. 51164 Protects Actin Filaments in Cytochalasin D Treated Fibroblasts

To elucidate whether 51164 could affect the changes in F-actin reorganization, we exposed cultured primary mouse fibroblasts to 2.5 μg/mL cytochalasin D together with either 1 μM 51164 or an equal volume of DMSO for 15 min. To visualize the reorganization of F-actin microfilaments, cells were stained with rhodamine-conjugated phalloidin. We used two methods to quantify the organization of F-actin microfilaments: (1) quantification of the change in the mean grey value of rhodamine-phalloidin fluorescence [43], which allowed us to evaluate the changes in the mass of F-actin; and (2) measuring the fractal dimension change by the box-counting method [19,21,44,45]. The measurement of the fractal dimension allowed us to quantify the structural changes in F-actin organization.

We found that the mean gray value of rhodamine-phalloidin labeled F-actin in the control group + DMSO was 4621 ± 177 a. u. (n = 86). Treatment of fibroblasts with compound 51164 for 15 min did not change the mean gray value statistically [3902 ± 302 a. u. (n = 42), $p = 0.1968$ in comparison to the control group, two-way ANOVA following Sidak's multiple comparisons test] (Figure 3B). Incubation of cells with cytochalasin D during the same time led to a statistically significant decrease in the mean gray value to the level of [2455 ± 110 a. u. (n = 105), $p < 0.0001$, two-way ANOVA following Sidak's multiple comparisons test]. Incubation of cells in the presence of cytochalasin D together with compound 51164 for 15 min led to the increase in fluorescence to 4385 ± 264 a. u. (n = 86) (Figure 3B). This result is statistically indistinguishable from the control groups ($p = 0.6438$, two-way ANOVA following Sidak's multiple comparisons test).

The difference in the fractal dimensions of rhodamine-phalloidin labeled F-actin between the control group in the presence of DMSO or in the presence of 1uM 51164 was not significant [1.63 ± 0.01 (n = 86) in the control group, versus the group of cells treated with 51164 1.66 ± 0.01 (n = 42), $p = 0.9369$, two-way ANOVA following Sidak's multiple comparisons test] (Figure 3C). Cytochalasin D statistically decreased the fractal dimension in the control group [from 1.63 ± 0.01 (n = 86) to 1.54 ± 0.02 (n = 105), $p < 0.0001$, two-way ANOVA following Sidak's multiple comparisons test]. Incubation cells in the presence of

cytochalasin D together with compound 51164 increased the fractal dimension up to the control levels [1.64 $\pm$ 0.01 (n = 86)]. The decrease in fractal dimension indicated a noticeable perturbation in F-actin organization of the cells incubated with cytochalasin D. Elevation of the fractal dimension level in the cytochalasin D + 51164 group suggests that the F-actin organization returned to normal (Figure 3C).

Therefore, two different methods of fluorescent image analyses revealed that compound 51164 impacts actin organization in vitro.

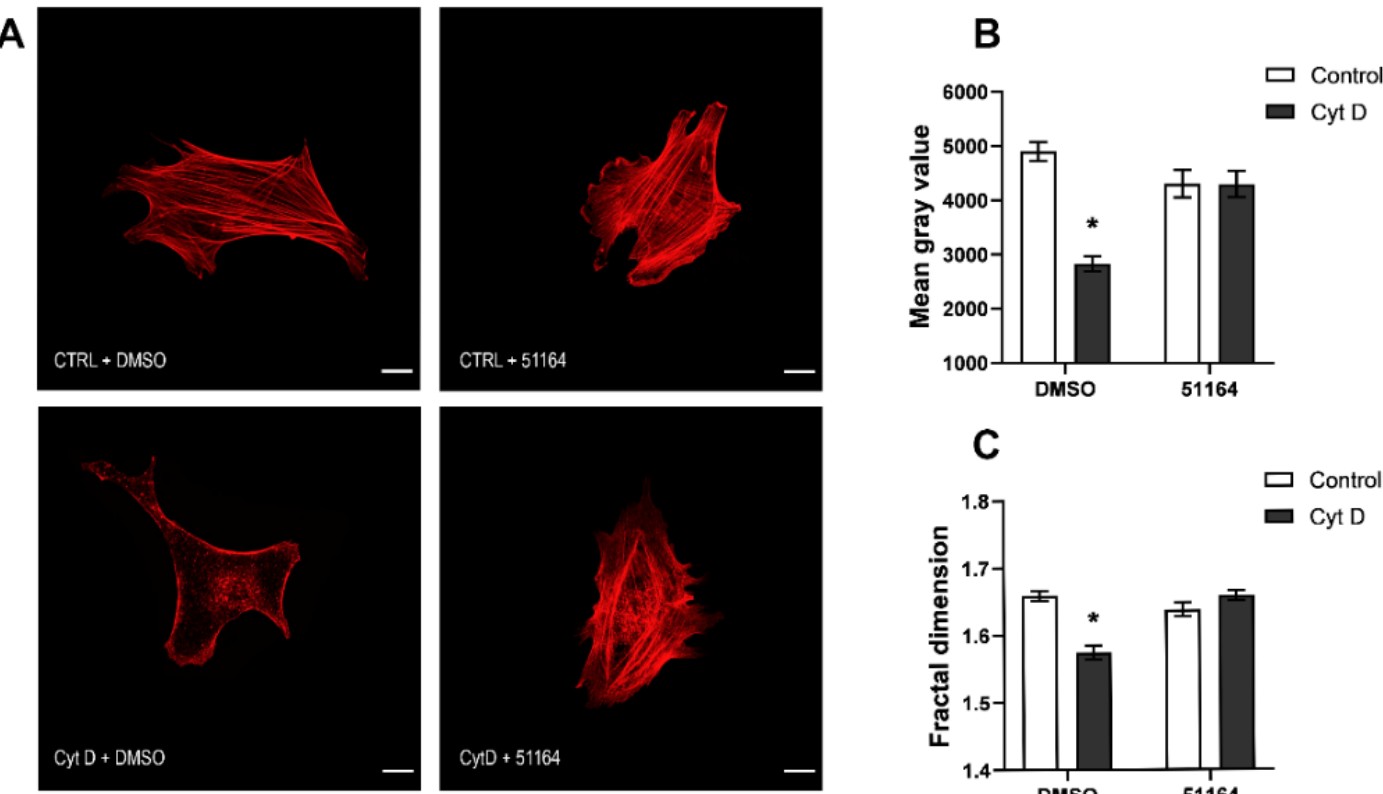

**Figure 3.** 51164 protects actin filaments in cytochalasin D-treated fibroblasts. (**A**) Representative confocal images of cultured primary mouse fibroblasts stained with rhodamine-phalloidin. The cells were incubated for 15 min in the absence (CTRL) or presence of cytochalasin D (Cyt D) and treated with 1 $\mu$M 51164 or an equal volume of DMSO. Scale bar = 25 $\mu$m. Bar charts of mean gray values (**B**) and fractal dimension mean values (**C**) were calculated for each group of cells. Results are presented as mean $\pm$ SEM, n (cells) = 15–35 per group from one experiment, the experiment was repeated three times. Normal distribution was checked by the Shapiro–Wilk test. Two-way ANOVA following Sidak's multiple comparisons test, * $p < 0.0001$.

*3.4. Molecular Docking Study*

Both the globular actin (G-actin) and its polymeric form, filamentous actin (F-actin), simultaneously exist in dendritic spines. The G-actin/F-actin ratio affects the spine morphology [46]. It was also discovered that long-term potentiation (LTP) induction shifts the G-actin/F-actin ratio toward the prevalence of F-actin and increases the spine's volume, whereas long-term depression (LTD) induction, in contrast, shifts the ratio toward G-actin and results in spine elimination [36]. To investigate the putative effect of 51164 on G-actin ability to form filamentous actin, we performed in silico studies including molecular docking and molecular dynamics. We took the jasplakinolide as a control compound that is well-known to interact and stabilize the actin structure. We tried to compare 51164 to jasplakinolide to characterize 51164 as potential actin binding and stabilizing agent. In the in silico studies, we used a crystal structure of $\alpha$-isoform G-actin. Today, it is a single native variant of G-actin available at the protein data bank.

The crystal structure of G-actin was first presented in [47] and consists of 375 amino acid residues with a molecular weight of 43 kD. This protein consists of α/β domains, known as outer and inner domains. It should be noted that the spatial dimensions of the domains are different. In the traditional sense of structural separation, these two domains are classified into four subdomains. Subdomains 1 and 3 are primary domains, and 2 and 4 are treated as inserts. Several key loops are present in this subdomain, which play an important role in the function of this protein. S-loop (residues 11–16) and G-loop (residues 154–161) are near the nucleotide-binding site and are involved in direct ATP/ADP binding. The shift in the G- and S-loops extends into the H-loop (residues 70–78), which contains an important H73 residue, whose methylation is considered an important activity that contributes to delaying the release of inorganic phosphates from actin subunits. The DNAse-I Binding Loop (D-loop, residues 38–52) is a disordered loop in G-actin and an ordered helix in F-actin. It is assumed that ADP/cofilin performs the function of actin depolymerization by directed trypsinization of loops 60–69. The W-loop (residues 165–172) is the binding site of profilin, cofilin, and twinfilin and plays a role in longitudinal and transverse actin interactions. The C-terminal (residues 349–375) and N-terminal (residues 1–10) loops are considered to be the regions of greatest structural flexibility [48].

Nowadays, more than 80 molecular models of G-actin are included in different databases. Many of them are presented in mutated forms or have an unstable active center. A native model with identification number PDB ID: 3HBT obtained by X-ray diffraction analysis was chosen [49].

To determine the possible binding sites of the studied ligands on the surface of G actin, "blind docking" was carried out. The results indicate that 51164 and jasplakinolide interact with G-actin, with different sites of interaction (Figure 4). 51164 interacts with the ATP binding site near the sensory loop, a jasplakinolide interacts near the C-loop, which plays an important role in actin polymerization [50]. The obtained conformational maps of complexation indicate that the 51164 interaction is of a mixed nature. Both electrostatic and hydrophobic forces were involved in the interaction, and the possible formation of a hydrogen bond with Lys336 with a distance of 2.68 Å was observed.

The interaction of jasplakinolide with the C-terminal-loop of G-actin is mainly electrostatic. Along with this, a hydrophobic type of binding to the amino acid residues Arg 116 and Tyr169 was also observed. The hydrogen type of binding of jasplakinolide to G-actin was observed with Ala170 and Lys373 in the range of 3.40. The energy values obtained for complexation at RMSD $\leq$2 Å for 51164 were −7.54 kcal/mol and for jasplakinolide -6.86 kcal/mol, respectively (Figure 4).

*3.5. Molecular Dynamics Study*

For a more detailed study of complex formation, based on the results of the spatial and energy characteristics obtained by molecular docking, a series of experiments were carried out using the molecular dynamics method. The steric and energetic characteristics of the complex formation of the studied ligands with G-actin were obtained. Based on the energy values, the binding constants for the studied complexes were calculated.

The spatial parameters of the complex formation of 51164 with G-actin obtained by us indicate that the interaction occurred in the ATP binding site. Two metastable hydrogen bonds were observed with Lys 336 and Gly156 with a distance of 1.65Å and 2.61Å, respectively. The main contribution to the hydrogen type of binding was on Lys 336 and the hydroxyl group in 51164. It is known that this amino acid is a stabilizing factor between subdomains 3 and 4 [51]. Amino acid residue Lys18 is involved in the complex formation: Met305; Thr303; Gly182; Asp157, which interact based on van der Waals forces (Figure 5).

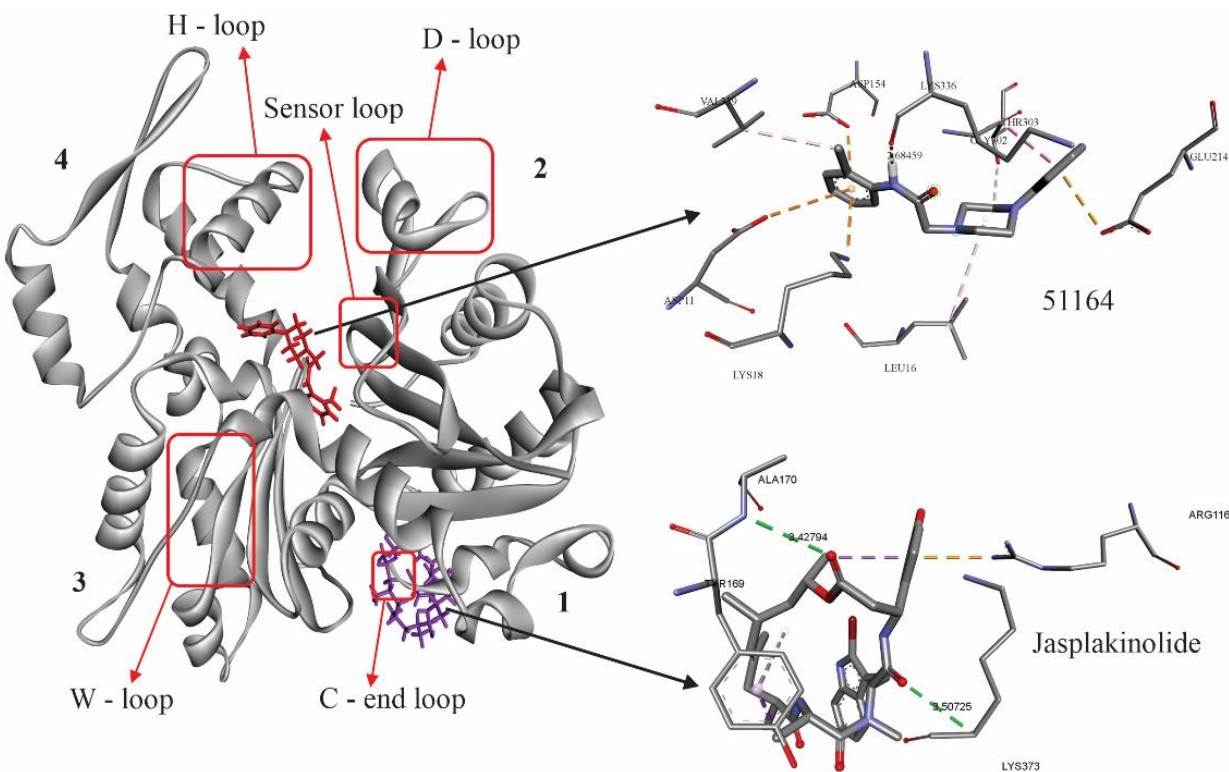

**Figure 4.** Spatial orientation 51164 (in red) and jasplakinolide (violet) on the G-actin surface, obtained by molecular docking. The key loops involved in the polymerization of G-actin are also indicated [50,51]. The numbers show the subdomains of the protein.

The 51164-G-actin complex also contains hydrophobic forces with the amino acid residues Leu16 and Val339. An electrostatic type of interaction was observed in Lys 213 and Glu214. The 51164 interaction does not directly affect the ATP catalytic triad (Asp11, Gln137, Asp154) [49]. At the same time, it is known that Asp157, together with Ser14, plays an important role in the process of ATP catalysis by conformational changes in the steric parameters of His 73 during methylation [51].

As predicted by molecular docking, the binding site of jasplakinolide differed from 51164. Jasplakinolide interacted with the C-terminal region near the W-loop (Figure 6). Hydrogen forces, as in the case of 51164, were not stable and covered the amino acid residues of Arg116, Lys373, Asn111, His371 with distances not exceeding 3.06 Å. Percentage-wise, relatively stable hydrogen bonding was observed in ARG 116 with a distance of 1.91Å over 5 ns, after which no hydrogen bonding was observed based on the trajectory change of jasplakinolide at the binding site. During the interaction, jasplakinolide changes its position in the direction from the C-terminus to subdomain 3 near the W-loop. The hydrophobic type of binding was observed in Pro172 and Tyr169. All other residues exhibited the van der Waals type of interaction. It should be noted that under these fluctuations, the interaction with Tyr169 and Phe375 was stable. These residues are known to play an important role in D-loop stabilization and promote G-actin polymerization [52].

The RMSD values obtained by us for the two complexes were stable throughout the simulation and did not exceed 1.3 nm. The obtained energy values of the interaction for the studied complexes indicated that the complex formation was mainly carried out due to electrostatic and van der Waals forces. The high value for both Coulomb and van der Waals forces was 51164. In terms of RMSD, when compared to jasplakinolide, 51164 also had a stable value. The results of the calculations of the total interaction energy indicate that 51164 binds more strongly to G-actin compared to jasplakinolide (Figure 7).

Based on the obtained interaction energies, the binding constant of the studied ligands during complexation was calculated (for 51164 Kb = 1.5 × $10^8$ and for jasplakinolide

1.37 × 10^6). Thus, the obtained spatial–energy characteristics of complex formation indicate that 51164 can lead to the stabilization of G-actin through the interaction of Lys 336 and Asp157. The stabilization factor 51164 is also indicated by the interaction with conjugated amino acid residues that form the ATP binding site. 51164 must not affect the ATP catalytic triad, which could lead to the competition when binding native ligand. This was also indicated by the types of forces involved in complex formation. From an energetic point of view, the calculated binding constants show that 51164 binds much more strongly to G-actin than jasplakinolide. Although 51164 differs in steric characteristics and binding sites from jasplakinolide, conformation analysis data demonstrated a hydrogen bond between the Lys336 residue of actin and the hydroxyl group of 51164 (Figure 5B). Lys336 is a stabilizing factor between subdomains 3 and 4 of G-actin [51,53]. Moreover, the interaction with Asp157 and Ser14 directly affects the conformational changes of His73 (Figure 5B,D), which play a key role in the stabilization/polymerization of the structure of G-actin and is involved in the process of ATP catalysis [51]. Thus, we speculate that 51164 may be characterized as a chemical that plays a role in the stabilization and polymerization of G-actin.

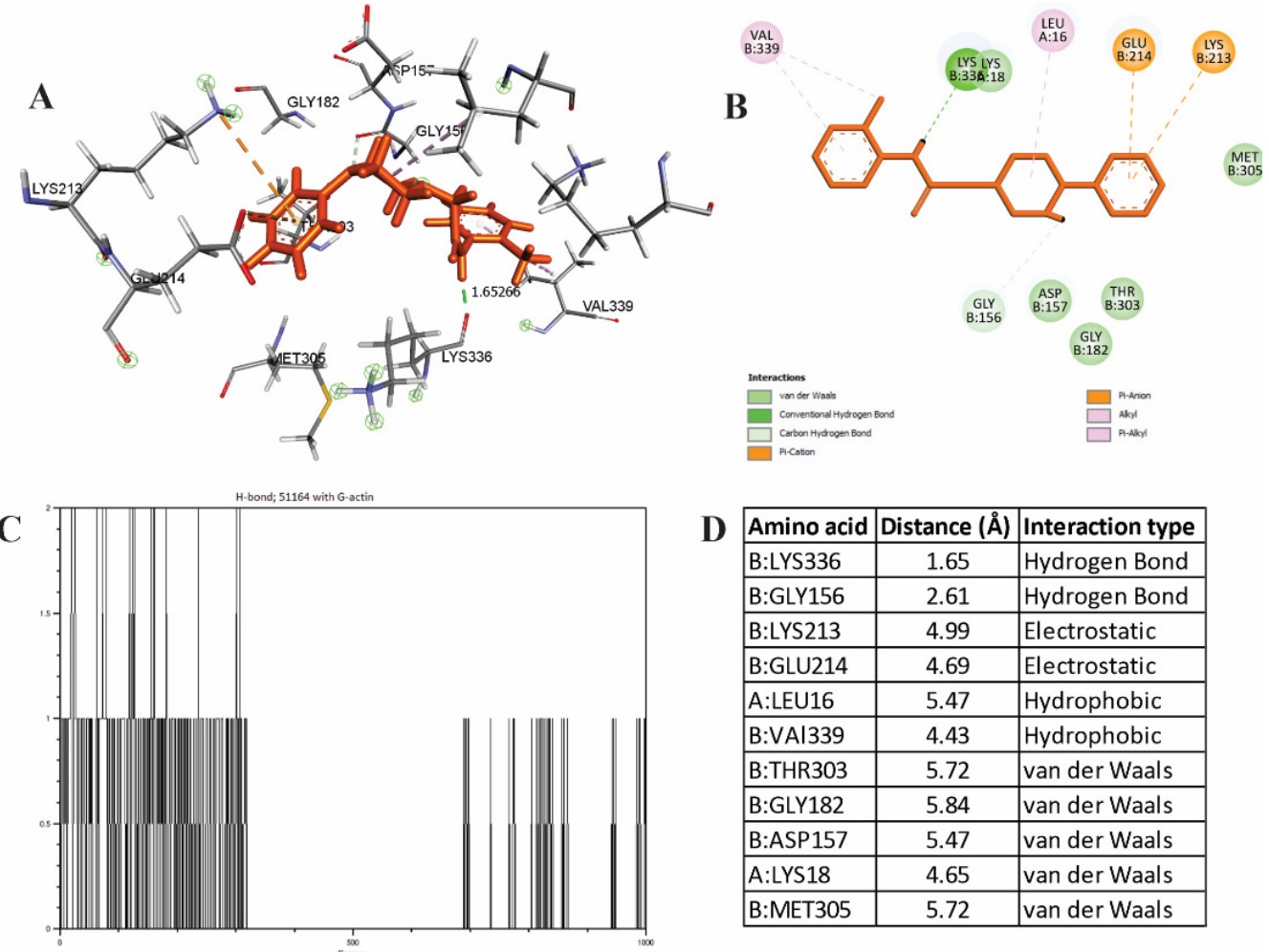

**Figure 5.** Spatial properties of 51164–G-actin complexation. (**A**) Spatial arrangement of 51164 in the ATP binding site. (**B**) Conformational map of the complexation of 51164 with G-actin. (**C**) The number of hydrogen bonds during complexation, (**D**) The list of amino acids involved in the interaction.

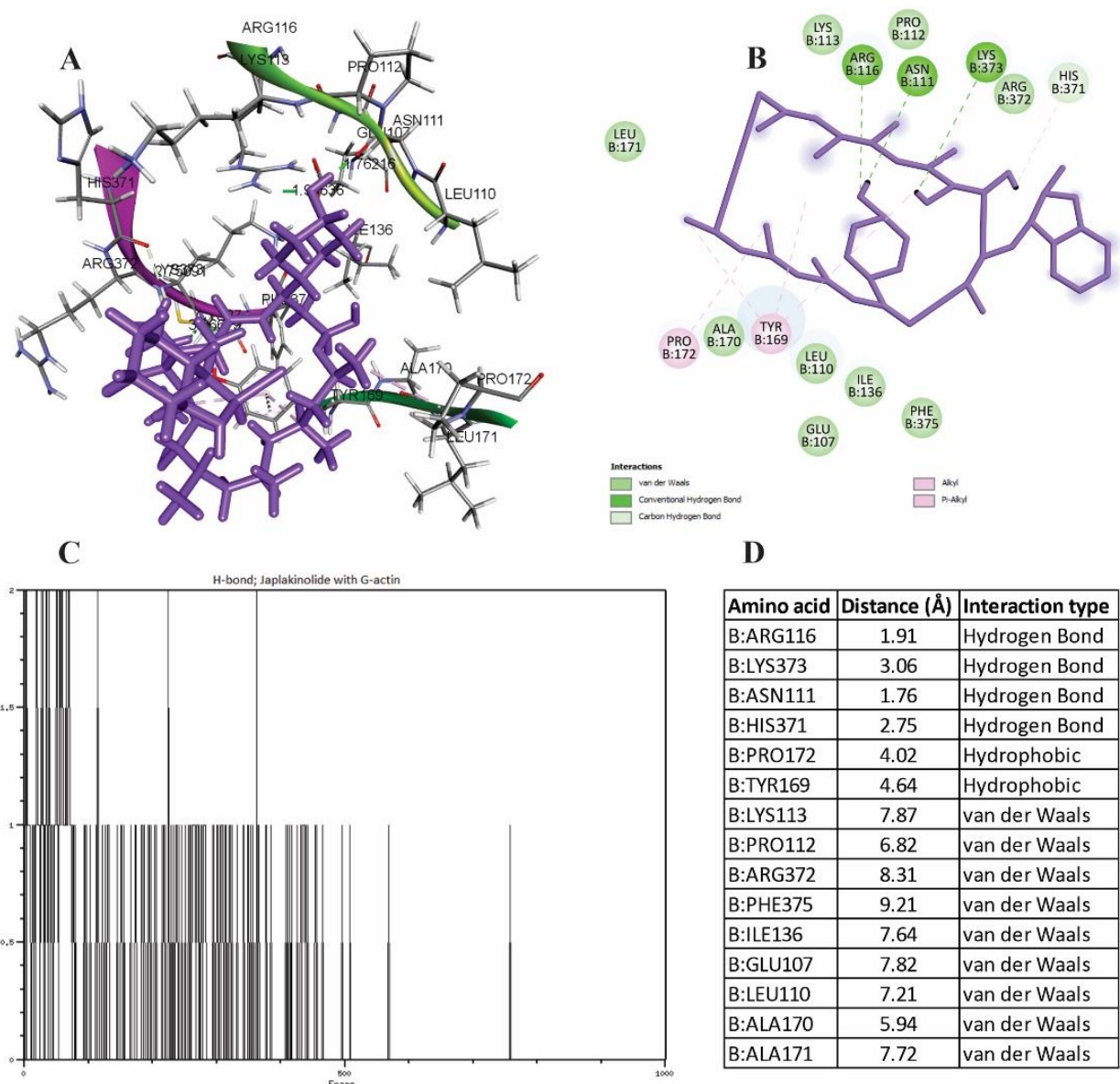

**Figure 6.** Spatial properties of jasplakinolide–G-actin complexation. (**A**) The spatial arrangement of jasplakinolide in the C-terminal loop of G-actin. (**B**) Conformational map of the complexation of jasplakinolide with G-actin. (**C**) The number of hydrogen bonds during complexation. (**D**) The list of amino acids involved in the interaction.

## 4. Discussion

The current paper describes an alternative mechanism of piperazine derivative, compound 51164. The mechanism is related to the stabilization and promotion of actin polymerization in vitro. Whether this mechanism is dominant or secondary for 51164 was unable to be dissected for now.

We observed that 51164 is able to maintain the mushroom spine percentage in the absence of CaMKIIβ in a manner that does not depend on TRPC6-mediated nSOCE in the hippocampal spines. Previously, the PASS online web service showed that 51164 is able to activate the voltage-sensitive calcium channel, sigma 1 receptor as well as neuropeptide Y2 [13]. In the current paper, we did not perform experiments to exclude these cross specificities.

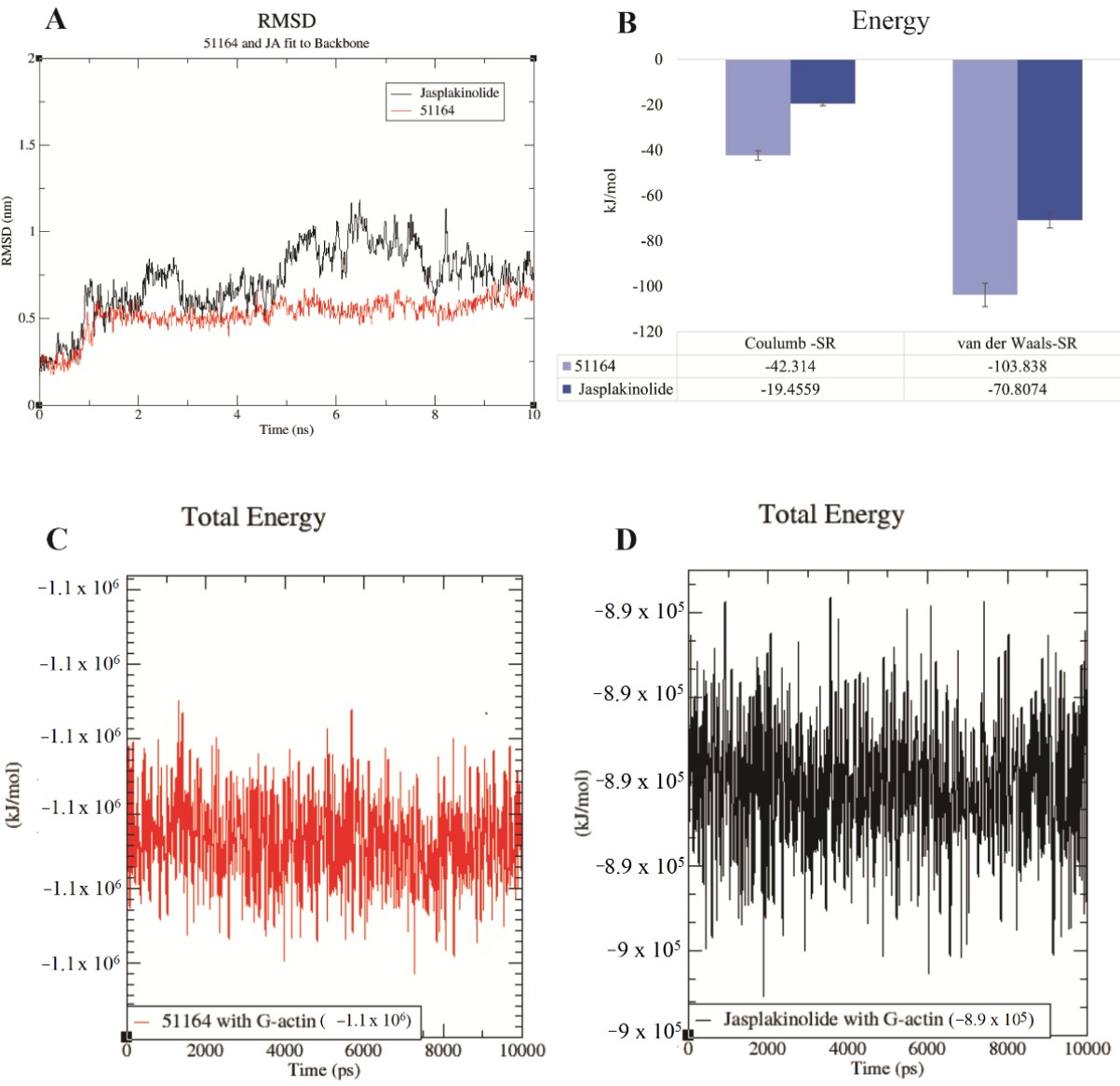

**Figure 7.** Calculated spatial–energy values for the complexes of 51164–G-actin and jasplakinolide–G-actin. (**A**) The RMSD value for the study complexes. (**B**) Calculated values of Coulomb and van der Waals forces for two complexes. Total energy values of 51164–G-actin (**C**) and jasplakinolide—G-actin (**D**).

However, we obtained strong in vitro and in silico data indicating that 51164 might play a role in the formation of F-actin. Since the effect of 51164 on F-actin formation was only observed in primary fibroblast, we skipped the discussion of 51164 as an actin stabilizing/polymerizing drug in neurons.

It is known that jasplakinolide interacts with F-actin, which is a polymeric form of G-actin, participating in its polymerization [54]. Along with this, there are data that jasplakinolide also acts with G-actin [55,56]. Our results showed that jasplakinolide interacts with amino acid residues in the C- and W-loops of G-actin. Both loops play an important role in the G-actin stabilization/polymerization process [50]. Based on the results obtained in silico, it was found that jasplakinolide interacts with the amino acid residues Lys373; His371, Arg372, and Phe375, which are part of the C-terminal loop of G-actin. In the case of 51164, the interaction occurs in the proximity of the ATP catalytic triad without affecting it. We showed that 51164 interacts with Lys 336, which is a stabilizing factor between subdomains 3 and 4 [51]. On the other hand, the interaction with Asp157 and Ser14 directly affects the conformational changes of His73, which plays a key role in the

stabilization/polymerization of the structure of G-actin and is involved in the process of ATP catalysis [51]. Based on the obtained data, it can be supposed that the involvement of the above-mentioned key amino acid residues in the complexation with 51164 leads to the stabilization of the G-actin structure. Despite different mechanisms of action, 51164 along with jasplakinolide can be considered as a positive modulator for G-actin, resulting in the stabilization/polymerization of its structure.

From a pharmacological point of view, it is not clear whether this property of 51164 to upregulate the actin polymerization would not cause severe side effects since no actin-targeting drugs have been used in clinical trials due to their cytotoxicity. One reason for this is that actin destabilizing drugs may disrupt filaments in both non-tumor and tumor cells [57,58]. Actin-stabilizing compounds also raise questions about their safety. Jasplakinolide had a narrow margin of safety in mouse cancer models and acute toxicity studies in rats and dogs. It was assumed that the cardiotoxicity caused an observed deleterious effect [59]. In addition, tolerated doses studies in zebrafish suggest that jasplakinolide does not have significant acute toxicity, but might have a toxic effect on long-term survival [60].

Another argument on whether the actin stabilizing/polymerizing effect of 51164 is dominant or not is the amount of 51164 molecules needed to stabilize one molecule of G-actin. Will one molecule of 51164 be enough to stabilize several G-actin molecules within one F-actin filament? Based on the obtained in silico data, we speculate that actin stabilization occurs when the G-actin:51164 ratio is 1:1. This means that there is a need to deliver quite high concentrations of 51164, almost the same molar amount as actin, which is usually difficult to achieve and if the achieved would cause severe side effects.

However, piperazines, as actin-stabilizing compounds, may serve as a foundation for the search and development of a new class of anti-cancer drugs, since actin stabilizers contribute to proliferation inhibition and the impairment in the migration of cancer cells [61].

The main limitation of the study is the impossibility to translate data obtained on primary fibroblasts to neurons since actin-dependent molecular mechanisms differ between these two types of cells. To prove that 51164 plays a similar role in the brain, further studies including neurons as an object are necessary.

## 5. Conclusions

Here, we present novel data on the possible protective effect of the piperazine derivative, the compound 51164. In vitro and in silico data clearly demonstrated that 51164 is able to impact the actin structure. Whether 51164 plays a similar role in neurons is an open question that needs further experimental support.

**Supplementary Materials:** The following supporting information can be downloaded at: https://www.mdpi.com/article/10.3390/cimb44110353/s1, Video S1: Jasplakinolide with G_actin (violet); Video S2: 51164 with G_actin (red).

**Author Contributions:** E.P., L.H., N.Z., V.G., and A.M. wrote and edited the manuscript; N.Z. and D.M. conducted in vitro experiments; V.G. and A.M. conducted the in silico experiments. All authors have read and agreed to the published version of the manuscript.

**Funding:** This research was partially funded by Russian Science Foundation Grant No. 20-75-10026 (Figures 1–3), by the Ministry of Science and Higher Education of the Russian Federation as part of the thematic work "Activities to support of efficiency of Russian–Armenian and Belorussian–Russian Universities" (Figures 4–6) and in part by the program "Best International Grant for PhD" of Peter the Great St. Petersburg Polytechnic University (Figure 7).

**Institutional Review Board Statement:** The study was conducted in accordance with the Declaration of Helsinki, and approved by the Bioethics Committee of the Peter the Great St. Petersburg Polytechnic University at St. Petersburg, Russia (protocol code 1 and date of approval: 24 January 2022).

**Informed Consent Statement:** Not applicable.

**Data Availability Statement:** The molecular model of G-actin was taken from www.UniProt.org (accessed on 10 September 2022), with KB identification number: P68135. Jasplakinolide was stored in the PubChem database (identification number CID: 6436289).

**Acknowledgments:** We thank I. Bezprozvanny for the constructive critics of the paper and Anastasiya Bolshakova for the administrative support.

**Conflicts of Interest:** The authors declare no conflict of interest.

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
