# Peer review of "Piperazine Derivative Stabilizes Actin Filaments in Primary Fibroblasts and Binds G-Actin In Silico"

_cimb, doi:10.3390/cimb44110353_

Round 1

Reviewer 1 Report

The manuscript presents the success story of explaining how the compound 51164 (N-(2-chlorophenyl)-2-(4-phenylpiperazine-1-yl) acetamide) could recover synaptic plasticity in amyloidogenic mouse models of Alzheimer’s disease. The authors also demonstrated in-silico how the compound bound and therefore stabilized G-actin, which could relate to the preservation of actin filaments in-vitro. The molecular docking studies were followed by molecular dynamics (MD) simulations for 10 ns. This is a common practice now, but the MD simulations could be considered as too short. Nevertheless, these MD simulations were accompanied by convincing in-vitro studies. The in-silico studies suggested some important residues that could be helpful in further drug design projects for Alzheimer's Disease therapy.

The research presented in the manuscript is interesting and could be published in CIMB. Nevertheless, there are some minor things that could be corrected during the next step of the publication of the article, e.g., (i) What is "TRPC6" in line 14? (ii) What is "51164" in line 44? and (iii) What is "SOCE" in line 44? 

Reviewer 2 Report

The authors of the paper "Piperazine derivative stabilizes actin filaments in primary fibroblasts and binds G-actin in silico" present an article putting together some experiments done with cultured neurons, fibroblasts and then a second part about the probable interactions of the active molecule, 51164, to G-actin interfering its protective effect due to the ability to stabilise F-actin. 

I don't have the expertise to criticize the second part of the article. It seems properly written and I can follow the reasoning, but I lack the knowledge to evaluate this part, so I am going to focalise my comments to the first part.

The most accurate part of the paper is the title, both statements seems to be true. But the development of the article itself is a quite different story. The abstract and paper art talking about Alzheimer's disease and the role of mushroom spines in memory storage. Then, they explain that to stabilise mushroom spines they need a calcium entry and a TRCP agonist, the active molecule, can restore this calcium entrance in a cultured AD model protecting it against the effects of beta-amyloid oligomers.

In this part, the authors did not explain what TRCP stands for: transient receptor potential canonical 6 (TRPC6), making it harder to the eventual reader to understand it, but also they use a lot of phrases that are inaccurate in my opinion, i.e.: "increases mushroom spine percentage " I would say, "it maintains". " Connections between two neurons occur through the axon of one neuron and the dendritic spine of  another neuron" Mostly true in the case of excitatory synapses, but it is not always true, and of course not in the case of inhibitory synapses. Etc. 

Then, they realised two independent experiments including neurons, instead of three, as they did with the fibroblasts. Two replicates should no be enough to conclude replicability, even if the stats say that the difference is significant. So, the n should be increase for publication.

In the first experiment involving neurons, I believe that the result could be true, CamKII is known to be necessary for the normal functioning of the synapse, but the images showed as examples seems to be bad example. Meanwhile for shControl we can see two nice looking pyramidal neurons, for shCamKII we observe what seems to be granule neurons or interneurons, so raising the question if the difference could come from this difference. I guess that there has no be a selection of the population and there is an heterogeneity of cases studied, but definitely not the best examples to show.

In line 260 there is a typo, it says -18,8%, when it means to be just 18,8. Also in figure 2, the authors conclude that "51164 demonstrates an alternative synapto-protective effect that does not depend on TRPC6-dependent nSOCE " what it is not true. 51164 maintains the Mushroom spine proportion, but if we analyse the actual spine density, probably we will observe a decrease of total spines, just that in neutrons treated with DMSO the percentage of MS has been decreased in favor of other immature forms. But both shCamKII, treated or not, seems to have less spines than the shControls. In the figure legend they will not even talk about keeping the percentage, they say: "51164 rescues mushroom spine loss in hippocampal neurons with CaMKIIβ knockdown. " Affirmation that it is flagrantly false. The compound keeps the proportion, or I would like to see the total values to support this affirmation.

But this two experiments were confusing to me. They were talking about AD, neuroprotection, so I had interpreted that the experiments were made in that context but shCamKII is not a model of AD. That should be clarified.  

Then they decided to study the effect of 51164 on F-actin reorganization on fibroblasts instead of doing it on the dendritic spine heads, doing some FRAP experiments for example. Fibroblasts have other ways to regulate actin and nom other structures, like stress fibres. So, the conclusions on fibroblasts are not transposable to the neutrons dendritic spines. Also, the difference in the n of the different cases is big, from 42 to 105. More than double.

As I said, I will not comment the in silicon part.

Finally the discussion really surprised me. They did not discuss about the neuroprotective effect, implication on AD treatments, they prefer to talk about migration of cancer cells. Probably true, but not related with the previous premises expose in the introduction.

So, even if the results could be true and the compound interact with actin, I cannot recommend the publication of the article in its present form.  

Round 2

Reviewer 2 Report

The authors have modified the text, clarifying, running down some interpretations and making it more comprensible. Also, they adapted a figure and added some supplementary information. 

Still, the number of experiments continue being 2, even if there are a big number of neurons analysed (>100). Since I think that there should be at least 3, but I understand the difficulties that the authors could face to do it a 3rd time. I let the final decision in the hands of the editor. He/she would decide if 2 is enough for the standards of the journal.
